# UNSUPERVISED ONE-TO-MANY IMAGE TRANSLATION

## ABSTRACT

We perform completely unsupervised one-sided image to image translation between a source domain $X$ and a target domain $Y$ such that we preserve relevant underlying shared semantics (e.g., class, size, shape, etc). In particular, we are interested in a more difficult case than those typically addressed in the literature, where the source and target are "far" enough that reconstruction-style or pixel-wise approaches fail. We argue that transferring (i.e., *translating*) said relevant information should involve both discarding source domain-specific information while incorporate target domain-specific information, the latter of which we model with a noisy prior distribution. In order to avoid the degenerate case where the generated samples are only explained by the prior distribution, we propose to minimize an estimate of the mutual information between the generated sample and the sample from the prior distribution. We discover that the architectural choices are an important factor to consider in order to preserve the shared semantic between $X$ and $Y$. We show state of the art results on the MNIST to SVHN task for unsupervised image to image translation.

## 1 INTRODUCTION

Unsupervised image to image translation is the task of learning a mapping from images in a source distribution $X$ to images in a target distribution $Y$ without the use of any extrinsic information that can match the two distributions (e.g., labels). Some works (Liu et al., 2017; Zhu et al., 2017; Benaim & Wolf, 2017; Almahairi et al., 2018) have proposed solutions to this problem using intrinsic properties of the transfer. In order to preserve the relevant semantics, a common approach in all of these methods is to use pixels-wise consistency metrics. For example, CycleGAN (Liu et al., 2017) proposes a cycle consistency-loss between the input image and its reconstruction. Moreover, we assert that using pixel-wise consistency is unreasonably strong for problems where the source and target domains vary in relevant spatial characteristics.

The general problem statement and our solution is depicted in Figure 1 and 2, respectively. The translator takes two inputs, one is the source input that we wish to translate and the second is the independent noise meant to model statistical variation of the target not present in the source. In order to ensure the output of this translator resembles the target, we train this whole model as the generator in a generative adversarial networks (GAN, Goodfellow et al., 2014).

An important aspect of this problem is precisely what is meant by translation. Information content is one way to think of this, but unfortunately this quantity doesn't distinguish between things we care about (e.g., salient qualities such as shape, size, class, color, etc) and things we don't (noise). Furthermore, these *semantics* can be case-driven and can dependent on the end-goal of designing a translator. We assert in this paper that structural assumptions must be incorporated into our model. This insight is nothing new, recent works on representation learning as well as numerous works from computer vision and generative models (Doersch et al., 2015; Pathak et al., 2016; Hjelm et al., 2018; Oord et al., 2018) all operate on this assumption. Therefore, we perform our transfer in a way that forces the maintenance of spatial characteristics across transfer (i.e., by architecture choice).

That said, in order to encourage transfer between the source and target domains, we use mutual information as an additional objective for the translator. While the mutual information is intractable to compute for continuous variable, Mutual Information Neural Estimation (MINE, Belghazi et al., 2018) showed that this is not only possible, but this same estimator can be used to provide a gradient signal for a generator. We observe that using MINE to either minimize (between the independent

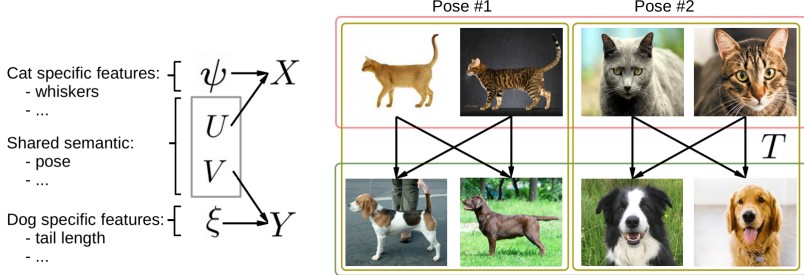

Figure 1: Formulation of the image to image translation problem. Given two domains $X$ and $Y$, presented as cat and dog images, we postulate that these two domains are in part explained by random variables $U, V$ from the shared semantic space and random variables $\psi$ and $\xi$ independent of $U, V$ that explain features specific to $X$ and $Y$, respectively.

noise and the output) or maximize (between the source and target variables) mutual information performs reasonably well on completely unsupervised tasks that models that rely on pixel-wise consistency performs poorly on, in our case MNSIT to SVHN, obtaining 71% accuracy on the transfer task.

The contribution of this paper are the following:

- We formalize the problem of unsupervised translation
- We propose an augmented GAN framework that takes two input and demonstrate competitive results on the image to image translation tasks
- We propose to use the mutual information to avoid the degenerate case where the generated images are only explained by one of the inputs by using the information theory

## 2 PROBLEM FORMULATION

In this section, we attempt to formalize the problem of unsupervised translation. The following can be seen as a minimal addition to the set of hypotheses usually assumed (overtly or not) in the machine learning literature.

Given distributions $\mathbb{P}_X$ and $\mathbb{P}_Y$ on the domains $\mathcal{X}$ and $\mathcal{Y}$, respectively, we assume that there exist random variables $U, V$ on a measurable space $(\mathcal{S}, \Sigma)$ such that,

(A) Given $U$ and $V$, $X$ and $Y$ are conditionally independent:

$$\mathbb{P}_{X|(U,V)} \otimes \mathbb{P}_{Y|(U,V)} = \mathbb{P}_{(X,Y)|(U,V)}$$

(B) Distributions $\mathbb{P}_{U|X}$ and $\mathbb{P}_{V|Y}$ are deterministic.

(C) $\operatorname{supp}(U) = \operatorname{supp}(V)$.

We would call $(\mathcal{S}, \Sigma)$ a *shared semantics* between $X$ and $Y$. In other words, we postulate that the observed data was generated by a directed probabilistic graphical model (PGM) shown in Figure 1 (i.e., would contain the directed edges $X \leftarrow U, V \rightarrow Y$). Note that this PGM is simply a conceptual tool to help us understand more deeply the task we are trying to solve.

It is important to ask why variables $U$ and $V$ need to be different. As an example, one can imagine that if $X$ and $Y$ represent two completely different visual representations of digits, say one from MNIST (LeCun & Cortes, 2010) and the other a uniformly drawn typeset digits with two possible fonts (e.g., *Helvetica* and *Times*). $Y$ is therefore a discrete variable with equally probable 20 different values. Intuitively, the shared semantics between $X$ and $Y$ is the notion of *digit*. Yet, the different classes in MNIST dataset are not evenly distributed. Therefore, the shared semantics cannot be modelled with a single random variable on the set $\{0, 1, \ldots, 9\}$. In fact, if we consider *any* variable $Y'$ with the support $\{0, \ldots, 9\} \times \{\text{Helvetica}, \text{Times}\}$, it's shared semantics with MNIST would

stay the same. Therefore, we model shared semantics through measurable space $(\mathcal{S}, \Sigma)$, while probabilities of certain modes in $\Sigma$ by separate variables $U$ and $V$, as they may differ between source and target variables.

Assumption (B) requires each of the variables $X$ and $Y$ to have a single representation (*semantics*) in the shared semantics space. Assumption (C) essentially says that these semantics are shared, i.e. for each set of semantics of $X$ with non-zero probability, there exists a set if samples of $Y$ with nonzero probability and the same semantics.

Given those assumptions, we can now define the general *Domain Transfer* problem. Given (possibly empirical) distributions $\mathbb{P}_X, \mathbb{P}_Y$ on domains $\mathcal{X}, \mathcal{Y}$ the learner attempts to find a (possibly) non-deterministic mapping $T : \mathcal{X} \to \mathcal{Y}$ and a shared semantics $(\mathcal{S}, \Sigma)$ such that,

1. $\operatorname{supp}(T(X)) = \operatorname{supp}(Y)$ (covering modes),
2. $\mathbb{P}_{U|X} \equiv \mathbb{P}_{U|T(X)}$ (shared semantics),
3. $\Sigma$ is *sufficiently large* (capacity).

Although the last assumption is somewhat vague in generality, if we fix sufficiently large $\mathcal{S} \subset \mathbb{R}^n$, the $\sigma$-algebras on $\mathcal{S}$ can be ordered by inclusion. We postulate that the quality of transfer depends on the expressiveness of the shared semantics $\Sigma$.

## 2.1 Individual features

Shared semantics assumption implies existence of features in $X$ that do not depend on $Y$ and vice versa. This directly related to *many-to-many* transfer: each sample from $X$ corresponds to a set of samples drawn from $\mathbb{P}_{Y|X}$ and each sample from $Y$ corresponds to many samples from $\mathbb{P}_{X|Y}$. Therefore, modelling a domain transfer from $\mathcal{X}$ to $\mathcal{Y}$ can be done with the use of additional variable $\xi \in \Xi$ that captures the variability of $Y|X$, so that there exists a deterministic map $G : \mathcal{S} \times \Xi \to \mathcal{Y}$. This idea has been employed before by Almahairi et al. (2018).

On the other hand, the individual features of $X$ that should stay independent of $Y$, need to be forgotten in the domain transfer, and the shared semantics should not include such features.

## 2.2 Inherent difficulty of learning a domain transfer

The fact that shared semantics between the translated domains cannot in general be expressed as a random variable causes inherent difficulties in training a transfer mapping. As respective modes of variables $X$ and $Y$ may have different probabilities, any mapping

$$T : \mathcal{X} \times \Xi \ni (x, \xi) \to y \in \mathcal{Y} \tag{1}$$

that is trained to match the distribution of $Y$ would suffer from the imbalance between frequencies of modes in the input and the target.

Although we do not propose a solution to this problem, we want to raise awareness of it in the research community. A possible solution of this problem would require a loss function of a generative model to penalize the distance of generated samples $y_i$ from the support of $Y$ independently of their likelihoods with respect to $\mathbb{P}_Y$

## 2.3 Ensuring the expressiveness of the shared semantics

The postulated *shared semantics* is not unique. In fact a trivial space $(\mathcal{S}, \Sigma) = (\{0\}, \{\{0\}\})$ satisfies assumptions (A)-(C), while a *random* transfer from $\mathcal{X}$ to $\mathcal{Y}$ satisfies *covering modes* and *shared semantics* conditions. Therefore, having a shared semantics space of sufficient capacity is crucial for the quality of the transfer.

Even with large enough space $\mathcal{S}$, learning a mapping 1 we bear a risk of it ignoring the actual source, inferring the $Y$ only from $\xi$. This is equivalent to trivial $\sigma$-algebra $\Sigma = \{\varnothing, \mathcal{S}\}$ and again leads to a random transfer.

We believe that one particular factor that may reduce this risk is the choice of the architecture of the $T$ network, as this architecture in fact models the shared semantics space $\mathcal{S}$. For instance, if $T$

is a neural network that takes two arguments, $x$ and $\xi$, the shared semantics lays in the last hidden layer representation of $x$ that do not depend on $\xi$. However, if the following layer ignores this representation, we again end up with trivial shared semantics.

Hence, we also propose to use *Mutual Information* (Belghazi et al., 2018) to reduce the above risk. In particular, one may encourage the network to keep dependence between $X$ and $Y$ by maximizing the MI between $X$ and $Y$. Alternatively, $T$ can be penalized for inferring $X$ from noise $\boldsymbol{\xi}$, by minimizing $I(X; \boldsymbol{\xi})$. We discuss both approaches in Section 4.2 and present empirical evidence of their efficacy.

### 2.4 PAIRED TRANSFER

As opposed to unsupervised domain transfer, in paired transfer, the learner has access to pairs $(x_i, y_i)$ $i = 1, ..., n$ independently sampled from the joint distribution $\mathbb{P}_{X,Y}$. Note that this setting can be reduced to supervised learning.

## 3 RELATED WORK

### 3.1 GENERATIVE ADVERSARIAL NETWORK

Generative Adversarial Networks (GANs; Goodfellow et al. (2014)) form a family of powerful implicit generative models that allow one to generate samples that mimic the target distribution, given only a sample from the latter. GANs have been the cornerstone of many advances in unsupervised learning and generative models.

In a GAN framework, two functions, a generator $G$ and a discriminator $D$, are trained simultaneously. The generator aims to generate sample in such a way that the discriminator cannot tell if the generated samples come from the true distribution. The discriminator is trained to distinguish the real samples from the generated samples. The orginal GAN objective function is defined as:

$$\min_G \max_D V(D, G) = \mathbb{E}_{x \sim \mathbb{P}_x} \left[ \log D(x) \right] + \mathbb{E}_{z \sim p_z(z)} \left[ \log(1 - D(G(z))) \right]. \tag{2}$$

Numerous variants and improvements of GAN's have since been proposed, including Wasserstein GAN (Arjovsky et al., 2017; Gulrajani et al., 2017), Spectral-Normalization GAN (Miyato et al., 2018), Conditional GAN (Mirza & Osindero, 2014).

### 3.2 IMAGE TO IMAGE TRANSLATION

CycleGAN (Zhu et al., 2017) adapts the GAN framework to domain transfer, learning two generators $G_Y : X \to Y$ and $G_X : Y \to X$ between domains $X$ and $Y$, and corresponding discriminators $D_X$ and $D_Y$. The transfer functions are trained together using combined loss that includes GAN objectives and a *Cycle Consistency Loss*. ALICE (Li et al., 2017) has shown that, in essence, this constraint is a lower-bound on the conditional entropy. However, maximizing the conditional entropy is not enough to assure alignment. Moreover, this consistency loss has side effects as shown in Chu et al. (2017) and prevents from learning a one-to-many mapping. Augmented CycleGAN (Almahairi et al., 2018) propose to transfer a pair of image and noise sample in order to learn a one-to-many mapping. Distance propose to preserve the L1 distance between each pairs of samples.

In the context of *shared semantics* defined in Section 2. CycleGAN through its deterministic character assumes that all semantics are shared between domains $\mathcal{X}$ and $\mathcal{Y}$, effectively setting $(\mathcal{S}, \Sigma) \equiv (\mathcal{Y}, \sigma(Y))$.

### 3.3 MUTUAL INFORMATION AND MINE

Mutual information is a natural way to think about information transfer in this setting, and recent models have leveraged estimating mutual information for learning deep representations (Hjelm et al., 2018; Oord et al., 2018). In information theory, the *Mutual Information* between two random vari-

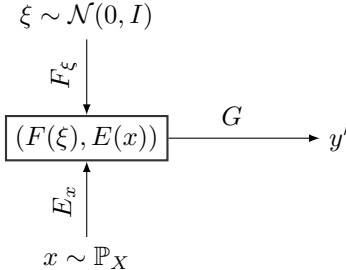

Figure 2: Computation graph of the *transfer network* $T$. First, normal noise $z$ is sampled and fed through $F$. We sample $x$ from the real data distribution $\mathbb{P}_X$ and fed through $E$. $F(\xi)$ and $E(x)$ are concatenated and fed through $G$ to get a generated sample $y'$.

ables $X$ and $Y$ with joint distribution $\mathbb{P}_{X,Y}$ is defined as,[1]

$$I(X;Y) \triangleq \underset{\mathbb{P}_{X,Y}}{\mathbb{E}} \left[ \log \frac{d\mathbb{P}_{X,Y}}{d(\mathbb{P}_X \times \mathbb{P}_Y)} \right]$$

where $\mathbb{P}_x, \mathbb{P}_y$ are marginal densities of $X$ and $Y$, respectively. Mutual Information can equivalently be expressed as $I(X;Y) = H(X) - H(X|Y)$, where $H(X)$ is the entropy of $X$ and $H(X|Y)$ is the conditional entropy of $X$ given $Y$. In essence, it measures the expected reduction in entropy of $X$ introduced by the observation of $Y$. Yet another useful formulation of the Mutual Information relates it to the Kullback-Leibler divergence,

$$I(X;Y) = D_{KL}(\mathbb{P}_{XY} || \mathbb{P}_X \otimes \mathbb{P}_Y)$$

between the joint distribution $\mathbb{P}_{XY}$ of $(X, Y)$ and the product of marginal distributions $\mathbb{P}_X \otimes \mathbb{P}_Y$.

In the usual GAN setting, none of those p.d.f. are tractable rendering exact computation and even MC estimation of the mutual information impossible. Since we can sample from those distributions, it is natural to transfer knowledge from the GAN literature to estimate this divergence.

That is exactly what Belghazi et al. (2018) propose with their *Mutual Information Neural Estimation* (MINE):

$$I_\Theta(X;Y) \triangleq \sup_{\theta \in \Theta} \underset{\mathbb{P}_{X,U}}{\mathbb{E}} [S_\theta(X,Y)] - \log \underset{\mathbb{P}_X \times \mathbb{P}_Y}{\mathbb{E}} [e^{S_\theta(X,Y)}] \qquad (3)$$

where $S_\theta$ is the *statistics network* which is a neural network parameterized by $\theta$. Belghazi et al. (2018) provides a proof that $I(X;Y) \geq I_\Theta(X;Y)$. In practice, the expectations on the r.h.s. of 3 are estimated via standard Monte Carlo sampling.

## 4    METHOD

In this section, we present our method for solving unpaired image to image translation as described in Section 2. We consider searching among a family of parametric mappings $\bar{\mathcal{T}} = \{T : \mathcal{X} \times \mathbb{R}^k \mapsto \mathcal{Y}\}$, such that $\bar{\mathcal{T}}$ is a family of all possible parameterizations of a given neural network architecture (see Figure 2). The process for generating *translated sample* $y'$ of an input $x$ we do the following:

1. First, sample $x \sim \mathbb{P}_X$
2. Sample $\xi \sim \boldsymbol{\xi} = \mathcal{N}(0, I)$
3. Feed $(x, \xi)$ through $T$ to get $y' = T(x, \xi)$.

Note that by searching into $\bar{\mathcal{T}}$, we are effectively searching in a s pace of stochastic mappings, namely $\mathcal{T} = \{T_Z | T_Z = T(\cdot, Z) \text{ for } T \in \bar{\mathcal{T}} \text{ and } Z \sim \mathcal{N}(0, I)\}$. Specifically, we model $T$ as the composition of networks $E$, $F$ and $G$, where

---

[1]Note here we assume that $\mathbb{P}_{X,Y}$ is absolutely continuous w.r.t. $\mathbb{P}_X \times \mathbb{P}_Y$, which is trivially true.

- $E : \mathcal{X} \rightarrow \mathcal{S}$ is modelled as a convolutional network and can be interpreted as an *encoder*,
- $F : \Xi \rightarrow I$ is a network composed of a linear layer and transposed convolutions,
- $G : (S, I) \rightarrow Y$ is a mapping from the concatenation of $S$ a $I$ to the target $Y$, and can be interpreted as a *decoder*.

We perform our search by using stochastic gradient descent and back-propagation with objectives described in the following sections.

## 4.1 ADVERSARIAL LOSS

In order to generate realistic looking samples in $Y$ we use an adversarial loss similar as the one presented in presented in Equation 2,

$$\min_G \max_D V(D, G) = \mathop{\mathbb{E}}_{y \sim \mathbb{P}_Y} [\log D(y)] + \mathop{\mathbb{E}}_{\xi \sim N(0, I), x \sim \mathbb{P}_X} [\log(1 - D(T(x, \xi)))] \tag{4}$$

## 4.2 MUTUAL INFORMATION ESTIMATION

As mentioned in Section 2.3, the transfer network can fall into a failure mode where $Y'$ is generated using $\Xi$ only. We have also observed that in some of our experiments. Figure A shows an example where the generated samples are only explained by the prior noise distribution. In order to avoid this possible failure case, we propose to use the estimated mutual information from Mutual Information Neural Estimation (MINE, Belghazi et al., 2018) to encourage $Y'$ to depend on $X$. In order to encourage the translator, $T$, to use the source, $X$, we have two choices: minimize the mutual information, $I(Z; T(X, Z))$, or maximize the mutual information $I(X; T(X, Z))$. The former is what is used in information bottleneck found in MINE, while the latter is found in their mode-regularizing experiments as well as recent papers on representation learning (Hjelm et al., 2018).

## 4.3 NETWORK ARCHITECTURE

The network architecture and the inductive bias associated to the networks decision is an important factor on the Image-to-Image translation. Given enough capacity, a network can find a transformation that satisfy our generation objective while not preserving the shared semantic. In other words, given two domains with the sames modalities. Given that theses modalities are not paired and unlabeled, it is impossible to guarentee a perfect mapping between each modalities. Therefore, we rely on the inductive bias coming from the choice of the architecture, constraining the family of function that the network can learn to functions that are close to the identity.

The transfer network and the discriminator are inspired from DCGAN. The statistic networks used to compute the mutual information are inspired from MINE. The downsamples are done using $4 \times 4$ kernels with stride 2. ReLU activations are used on the transfer network and on the discriminator. ELU activations are used on the statistics network. Spectral Normalization is used in the statistic network and the discriminator to stabilize the training.

We also carried out additional experiments using U-Net architecture (Ronneberger et al., 2015); details are presented in Appendix A.

## 4.4 TRAINING DETAILS

We use the non-saturating GAN loss as described in (Goodfellow et al., 2014). Other than the spectral normalization mentioned in the previous section, we do not use any other stabilizing technique. We use Adam with a learning rate of 0.0001 and default values of other hyperparameters for all the experiments. The factor of the mutual information penalty depends on the task.

## 5 EXPERIMENTS

In this section, we study two types of Image to Image translation. The first one consist of translation where the alignment does not require any geometric changes. We pick the edge to shoes dataset to study this type of translation and show that only a GAN loss is necessary to preserve the alignment

– anything else is unnecessary for this task. The second type of translation that we present consist of an alignment that requires geometric changes. To our best knowledge, none of the unsupervised image to image translation have been able to tackle this task. In order to evaluate this type of domain adaptation, we will look at the MNIST to SVHN task.

### 5.1 EDGE TO SHOES

Edge to shoes is a dataset of paired edges and shoes pictures. The training set is composed of 50000 pairs of edges and shoes. We do not consider the pairing when we are training or validating. We use images of dimension of $64 \times 64$.

Figure 3 and figure 4 compare the qualitative results on the edges to shoes dataset. We compare TI-GANwith and without the MI penalty with CycleGAN. CycleGAN has been trained using the code from the authors. We see that clearly, we can achieve good qualitative results using only the GAN loss. Each row represent a source that we wish to transfer. Each column is a different samples from the prior distribution. Because CycleGAN is a one-to-one mapping, we present only one result for this model.

In this task, the shoes domain is the same as the edge domain, but colored. Hence, the network can learn the identity mapping and learn to generate the colors using the information from $Z$. This is what we observe from the results from Figure 4. We can see that the MI penalty does not serve a purpose for this task. The GAN objective alone is enough to perform the transfer. Moreover, it looks like the cycle consistency blurs the quality of the sample. This is most likely the effect of the $l1$ loss.

When looking at Figure 3, we see that the noise $Z$ has no effect with our model, because the mapping is many-to-one. Again, only a GAN loss was necessary to achieve such results.

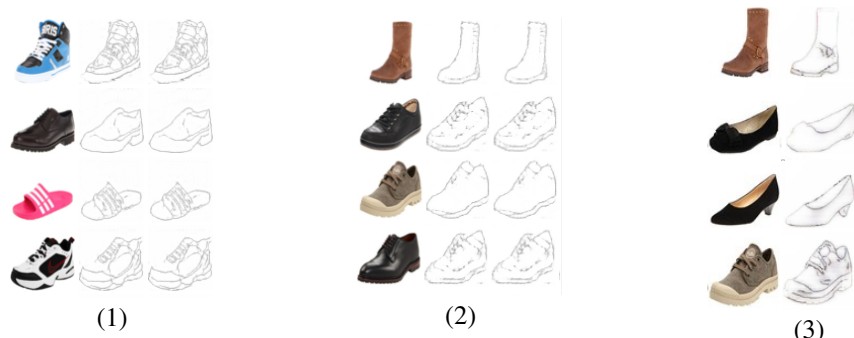

(1)      (2)      (3)

Figure 3: $64 \times 64$ Edge to shoes samples generated using (1) TI-GANmaximizing $I(\hat{X}; Y')$, (2) Only a GAN penalty, (3) Using CycleGAN recommended architecture

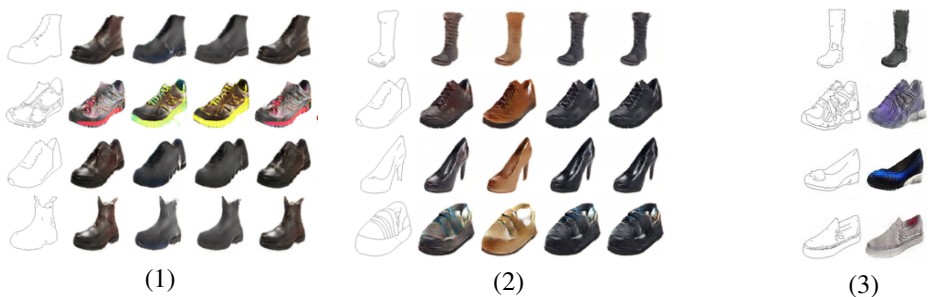

(1)      (2)      (3)

Figure 4: $64 \times 64$ Edge to shoes samples generated using (1) TI-GANmaximizing $I(X; T(X, \Xi))$, (2) Only a GAN penalty, (3) TI-GANusing CycleGAN recommended architecture

## 5.2 MNIST TO SVHN

MNIST to SVHN is a harder task because it requires a geometric transformation. For example, the typography of the SVHN digits is different from the MNIST digits, not all SVHN digits are centered, etc. Hence, learning a function close to the identiy mapping will not yield a good transfer. To our knowledge, no technique was able to perform well on this task.

Figure 5 compares the qualitative results on the MNIST to SVHN task from our method with and without the mutual information objective. Each row represents a MNIST digit that we wish to transfer. Each column use the same sample from a prior distribution. Samples in (1) are generated using only a GAN loss. Samples in (2) uses a GAN loss while minimizing $I(\Xi; T(X, \Xi))$. The shared semantic is well transferred and the independent factors of variations are explained by the prior distribution for the two experiments.

We can see SVHN to MNIST on figure 8. In this case, we noticed that regularizing the network using the mutual information, as described in section 4.2, helped to prevent this failure case.

Table 1 compares the transfer accuracy on MNIST to SVHN and on SVHN to MNIST. The transfer accuracy is obtained when evaluating the accuracy of the transferred samples using the labels from the source doomain and a classifier pre-trained on the target dataset. For this task, we pre-trained the classifiers using the architecture proposed in Simonyan & Zisserman (2014). We achieve an accuracy of 95.56% on the SVHN classification task and 99.6% on the MNIST classification task. We see an improvement in accuracy when using TI-GAN when minimizing $I(\Xi; Y')$. Overall, we see an important improvement of using our technique against CycleGAN. However, we note a lower accuracy on the SVHN to MNIST transfer task. We hypothetize that this is because even if we force the transfered MNIST samples to be dependent on SVHN, the translation networks has other factor of variation that it can pick to generate the MNIST digit (e.g. the color).


(1)
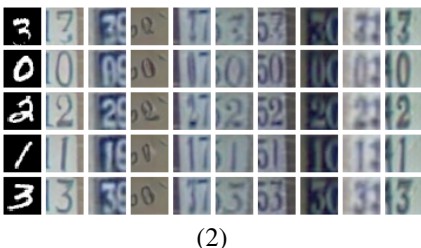
(2)

Figure 5: MNIST to SVHN using TI-GAN. The first column represent the source images to transfer. Each subsequent columns are different samples from the prior distribution used to transfer. (1) is our technique using only the GAN objective. (2) is our technique using the GAN objective while minimizing the mutual information of the generated samples and the samples from the prior

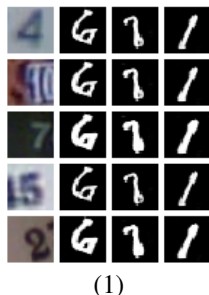
(1)
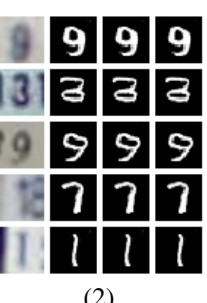
(2)

Figure 6: SVHN to MNIST using TI-GAN. The first column represent the source images to transfer. Each subsequent columns are different samples from the prior distribution used to transfer. (1) is our technique using only the GAN objective. (2) is our technique using the GAN objective while minimizing the mutual information of the generated samples and the samples from the prior

Table 1: Transfer accuracy on MNIST to SVHN and SVHN. The accuracy is obtained by using a classifier trained only using the sample of the target dataset. Evaluation is performed using the validation set of source dataset transfered to the target dataset.

| Method | SVHN (%) | MNIST (%) |
|---|---|---|
| CycleGAN | 19.6 | 21.2 |
| Ours + min $I(\Xi; T(X, \Xi))$ | 71.87 | 38.4 |
| Ours | 63.6 | 11.56 |

For MNIST to SVHN translation task, we carried out additional experiments using U-Net architecture (Ronneberger et al., 2015) with WGAN-GP (Gulrajani et al., 2017) as a noise function for the generative model. Results from these experiments are discussed in Appendix A.

## 6 CONCLUSION

In this paper, we present an unsupervised image to image translation framework. We formalize the problem of domain translation. We show that one to many image translation can be achieved without using any consistency loss. We explore the more complicated translation task where geometric changes are needed with the MNIST to SVHN task. We notice that the SVHN to MNIST task is harder. We hypothetize that this is due to the fact that SVHN has more factors of variation that the network can pick to generate the MNIST digits.

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

## A  RESULTS WITH UNET ARCHITECTURE

We carried out experiments with UNet-like architecture (Ronneberger et al., 2015) with depth 2 or 3, i.e. the skip connections were present at representations of size $(64 \times 16 \times 16)$ and $(128 \times 8 \times 8)$, and $(2564 \times 4)$ in the latter case. The results were generally worse than with the default architecture explained earlier, (we obtained accuracy of up to 45%) but we still believe these results are worth sharing. For instance, impact of Mutual information on the learnt model is very clear; in addition our model still performs much better than Cycle GAN with the same architecture.

All TI-GAN UNet models were trained with WGAN-GP (Gulrajani et al., 2017) and penalization of the mutual information between the noise and the output (i.e. min-max MINE setting).

Figures A and A present samples drawn from different UNet models.

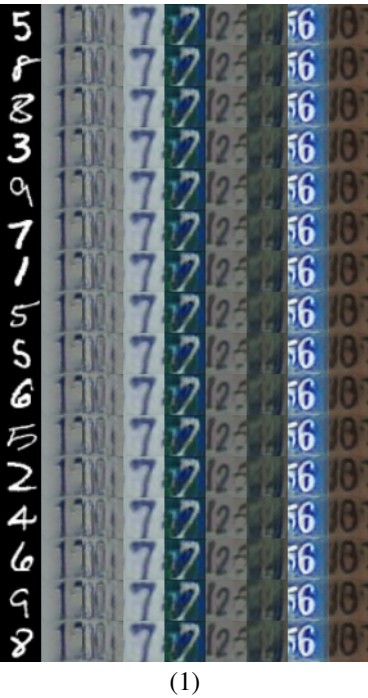   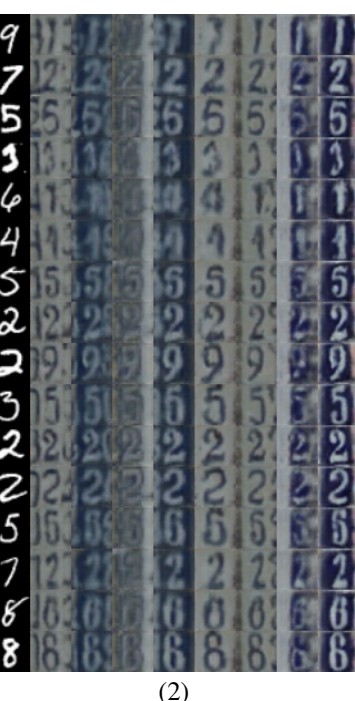

(1)                                    (2)

Figure 7: SVHN to MNIST using TI-GANwith UNet architecture. The first column represent the source images to transfer. Each subsequent columns are different samples from the prior distribution used to transfer. (1) is our technique using only the WGAN-GP objective. (2) is our technique using the WGAN-GP objective while minimizing the mutual information of the generated samples and the samples from the prior

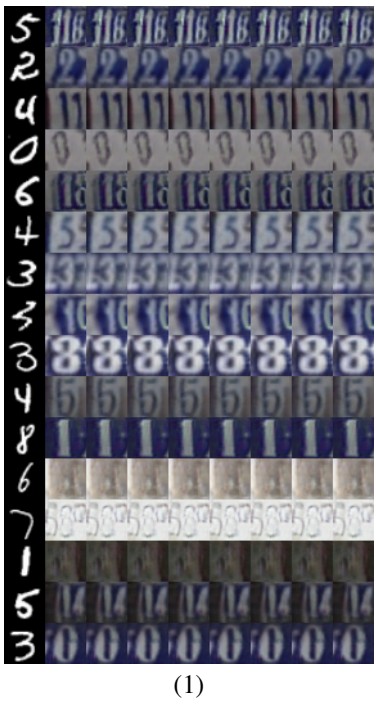

(1)

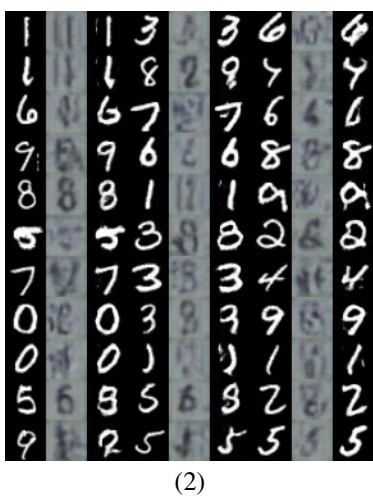

(2)

Figure 8: (1) SVHN to MNIST using TI-GANwith UNet architecture with higher weight of MI penalty. The first column represent the source images to transfer. Each subsequent columns are different samples from the prior distribution used to transfer. The higher MI penalty results with the network completely neglecting the noise and inferring all SVHN features from MNIST. (2) Cycle GAN with UNet architecture. Each row and three consecutive columns represent MNIST image, translation to SVHN and MNIST reconstruction.

