# OpenReview forum: "Unsupervised  one-to-many image translation"
_ICLR.cc/2019/Conference_

### Official Review · AnonReviewer2 · 2018-10-25
**Nice problem formulation but limited model novelty and comparisons.**

**Rating:** 4
**Confidence:** 4

**Review:**

This paper formalizes the problem of unsupervised translation and proposes an augmented GAN framework which uses the mutual information to avoid the degenerate case.

Pros:
* The formulation for the problem of unsupervised translation is insightful.
* The  paper is well written and easy to follow.

Cons:
* The contribution to the GAN model of this paper is to add the mutual information penalty (MINE, Belghazi et al., 2018) to the GAN loss, which seems incremental. I also wonder if some perceptual losses or latent code regression constraint used in previous works [1,2] can also achieve the same goal.
* Comparison to “Augmented CycleGAN: Learning Many-to-Many Mappings from Unpaired Data” should be done, since it’s a close related work for unsupervised many-to-many image translation.
* The visualization results of TI-GAN, TI-GAN+minI, CycleGAN should be listed with the same source input for fair and easy comparison. For example the failure case of figure 8 mentioned in Section 5.2 only appears in Figure 5 (1) not in Figure 5 (2).
* Minor issues: 1) What does the full name of “TI-GAN” ? 2) Figure 6 is not mentioned in the experiments. 3) What does the “Figure A” mean in Section 4.2 ?

[1] Multimodal Unsupervised Image-to-Image Translation, ECCV’18
[2] Diverse Image-to-Image Translation via Disentangled Representations, ECCV’18

Overall, this paper proposes a nice formulation for the problem of unsupervised translation. But the contribution to the GAN model seems incremental and comparisons to other methods are not enough. My initial rating is rejection.

---

> ### Author Response · Authors · 2018-11-24
> **Thank you for a detailed review.**
>
> Thank you AnonReviewer2 for the review. We refer to the lack of comparison in a general comment.
>
> > The visualization results of TI-GAN, TI-GAN+minI, CycleGAN should be listed with the same source input for fair and easy comparison. For example the failure case of figure 8 mentioned in Section 5.2 only appears in Figure 5 (1) not in Figure 5 (2).
>
> Good point, we will add that. However, we think that the results would reflect the same conclusion, that is Ti-GAN using a U-net architecture fails without the MI penalty.
>
> > What does the full name of “TI-GAN” ?
>
> Two-Input GAN. We will make it more explicit in the paper.
>
> > Figure 6 is not mentioned in the experiments.
>
> It should be mentioned, but due to a typo in the latex, we referenced figure 8 instead of figure 6. It will be fixed.
>
> > What does the “Figure A” mean in Section 4.2 ?
>
> We meant to reference the figures in the appendix A. We will make it more explicit by referencing the figures directly.

---

> > ### Comment · AnonReviewer2 · 2018-11-30
> > **Rating unchanged**
> >
> > Thanks for your rebuttal. Some issues are fixed but the comparisons with some other works, e.g. perceptual losses, latent code regression constraint and Augmented CycleGAN, are not mentioned. I still think the novelty and comparisons are limited. So I keep the rating.

---

### Official Review · AnonReviewer3 · 2018-10-26
**Good problem formulation, Not Novel method.**

**Rating:** 4
**Confidence:** 4

**Review:**

This paper formulated the problem of unsupervised one-to-many image translation and addressed the problem by minimizing  the mutual information. A principle formulation of such problem is quite interesting. However, the novelty of this paper is limited. The proposed the method is a simple extension of InfoGAN, applied to image-to-image translation and replacing the mutual information part with MINE.

The experiments, which only include edge to shoes and MNIST to SVHN, are also not comprehensive and convincing. This paper also lacks discussion of several quite important related references for one-to-many image translation.

XOGAN: One-to-Many Unsupervised Image-to-Image Translation
Toward Multimodal Image-to-Image Translation

---

> ### Author Response · Authors · 2018-11-24
> **Reply to Reviewer 3. Our motivation is different from InfoGAN.**
>
> Thank you AnonReviewer3 for the review. Lack of comparison is answered in a general comment.
>
> > The proposed method is a simple extension of InfoGAN, applied to image-to-image translation and replacing the mutual information part with MINE.
> The purpose of using the mutual information in our paper is different from the one presented in InfoGAN. In our paper, we use the mutual information as a mean to penalize the model for uniquely using the information coming from the noise distribution and disregarding the source.
> All I2I modes that aim to produce multimodal (many-to-many) transfer use some sort of prior noise to account for domain-specific features of the target domain (i.e. features not present in the source domain). This, however may lead to a failure mode, where learnt transfer function is agnostic to the source, as shown in Figure 7 (1).

---

### Official Review · AnonReviewer1 · 2018-11-02
**Good formulation, but not novel and short comparison**

**Rating:** 3
**Confidence:** 4

**Review:**

==== After rebuttal ===
I thank the authors for responses. I carefully read the response. But it is difficult to find a reason to increase the score. So, I keep my score.
====================

Unsupervised image-to-image (I2I) translation is an important issue due to various applications and it is still challenging when applied to diverse image data and data where domain gap is large. This paper employs a neural mutual information estimator (MINE) to deal with I2I translation between two domains where there is a large gap. However, this paper contains several issues.
1. Pros. and Cons.
   (+) Mathematical definition of I2I translation
   (+) Application of mutual information for conserving content.
   (-) Lack of comparison with recent I2I models
   (-) Lack of experimental results and ablation studies
   (-) Unclear novelty
2. Major comments
   - The novelty of this paper is not clear. Excluding the mathematical definition, it seems that the proposed TI simply combines DCGAN and MINE-based statistical networks. For clarifying the novelty, the detailed architecture and final objective functions can be helpful.
   - Recent works on unsupervised I2I translation are omitted including UNIT [1], MUNIT [2], and DRIP [3]. Also, the authors need to clarify the main difference of TI-GAN from comparing models.
   - It is not clear to relate the mathematical definition of domain transfer to one-to-many translation within large domain gap.
   - It is not clear how to use mutual information (MINE) for learning. There is no explicit definition of loss function considering MINE term.
   - It is short of comparing other state-of-the art models such as UNIT, MUNIT, DRIP, and AugCycleGAN. They compared their results with CycleGAN only.
   - Experiments are not enough to support the authors’ insist. There is not any quantitative metric or qualitative result on generating edge-to-shoes.
   - It is difficult to read due to inconsistent usage of terms (e.g., Figure 3 and 4 (c)s)
   - For better understanding, it requires to compare the patterns of MINE loss and adversarial loss.
   - Experiments on more datasets such as animal, season, faces or USPS datasets.
   - What is the main difference in the results between DCGAN-based and UNet-based models?


Minor
   - cicle_times symbol looks the product between distribution. But it should be defined before being used.
   - A reference of CycleGAN is incorrectly cited.
   - There are some typos in the paper.
   - page 1: dependent → depend
   - page 3: by separate → by separating
   - page 6: S a I → S and I


1. Ming-Yu Liu, Thomas Breuel, and Jan Kautz. Unsupervised image-to-image translation networks, CoRR, abs/1703.00848, 2017
2. Xun Huang, Ming-Yu Liu, Serge Belongie, Jan Kautz, Multimodal Unsupervised Image-to-Image Translation, CoRR. abs/1804.04732
3. Hsin-Ying Lee, Hung-Yu Tseng, Jia-Bin Huang, Maneesh Kumar Singh, Ming-Hsuan Yang, Diverse Image-to-Image Translation via Disentangled Representations, ECCV 2018.

---

> ### Author Response · Authors · 2018-11-24
> **Thank you for detailed review and pointing out important flaws.**
>
> Thank you AnonReviewer1 for the review and bringing some important points. Lack of comparison to existing models is answered in a general comment.
>
> > It is not clear how to use mutual information (MINE) for learning. There is no explicit definition of loss function considering MINE term.
> The total generator loss combines GAN loss and MI; the latter is estimated between noise prior and generated sample. MINE is optimized concurrently  to the GAN discriminator. We agree that explicitly stating TI-GAN objective is important; we will add it to the next revision of our paper.
>
> > It is difficult to read due to inconsistent usage of terms (e.g., Figure 3 and 4 (c)s)
> We will fix the inconsistency on Figures 3 and 4 that you pointed out
>
> > For better understanding, it requires to compare the patterns of MINE loss and adversarial loss.
> This is a good point. We will add a more thorough analysis of both MINE losses, which includes ablations studies and plots that evaluate the losses of each MINE estimator.
>
> > What is the main difference in the results between DCGAN-based and UNet-based models?
> UNet models achieved relatively good sample quality and disentanglement between semantics and SVHN-specific features. However, in comparison to DCGAN, the transfer was often incorrect. We will include qualitative results comparing the two architectures in the next version of our paper.
>
> > Minor comments
> We will add all your recommendation in the next version of our paper.

---

### Author Response · Authors · 2018-11-24
**General comment**

We would like to thank all reviewers for their effort and pointing out important flaws of the paper. We agree that more comparisons with more recent I2I translation technique are needed. In particular, more in-depth study of the previous work would make it clearer that certain I2I tasks, especially ones that involve more geometric changes (such as MNIST to SVHN), are not yet solved, and that the proposed model addresses certain problems involved in such tasks in a novel way.

---

### Meta-Review · Area_Chair1 · 2018-12-12
**Lack of novelty**

**Confidence:** 4
**Recommendation:** Reject

**Metareview:**

The paper formulates the problem of unsupervised one-to-many image translation and addresses the problem by minimizing  the mutual information.

The reviewers and AC note the critical limitation of novelty and comparison of this paper to meet the high standard of ICLR.

AC decided that the authors need more works to publish.